# SIFBench: An Extensive Benchmark for 3D Fatigue Analysis

## Abstract

Fatigue-induced crack growth is a leading cause of structural failure across critical industries such as aerospace, civil engineering, automotive, and energy. Accurate prediction of stress intensity factors (SIFs) — the key parameters governing crack propagation in linear elastic fracture mechanics — is essential for assessing fatigue life and ensuring structural integrity. While machine learning (ML) has shown great promise in SIF prediction, its advancement has been severely limited by the lack of rich, transparent, well-organized, and high-quality datasets.

To address this gap, we introduce SIFBench, an open-source, large-scale benchmark database designed to support ML-based SIF prediction. SIFBench contains over 5 million 3D crack and component geometries derived from high-fidelity finite element simulations across 37 distinct scenarios, and provides a unified Python interface for data access and customization. We report baseline results using a range of popular ML models — including random forests, support vector machines, feedforward neural networks, and Fourier neural operators — alongside comprehensive evaluation metrics and template code for model training, validation, and assessment. By offering a standardized and scalable resource, SIFBench substantially lowers the entry barrier and fosters the development and application of ML methods in damage tolerance design and predictive maintenance.

## 1 Motivation

Fatigue is a leading cause of failure in engineering structures across numerous critical industries, such as aerospace (Aliabadi et al., 1987), civil engineering (Albrecht & Yamada, 1977), automotive (Dehning et al., 2017), and energy (Callister Jr & Rethwisch, 2020). Components subjected to repeated or cyclic loading can develop cracks that propagate over time, even when the applied stresses are below the material's yield strength. This process, known as *fatigue crack growth*, can lead to sudden and catastrophic failure without prior warning. Ensuring the safety, reliability, and longevity of such structures hinges critically on the ability to accurately predict and manage the initiation and propagation of fatigue cracks.

The central parameter for predicting fatigue behavior is the stress intensity factor (SIF), a fundamental quantity in linear elastic fracture mechanics that describes the stress and strain fields near a crack tip (Newman, 2000). The SIF quantifies the severity of the stress concentration and is directly linked to the crack driving force (Irwin, 1957). It also determines the fracture mode — Mode I (opening), Mode II (in-plane shear), or Mode III (out-of-plane shear) — and, in the context of fatigue, serves as the primary variable governing crack growth under cyclic loading, as described by Paris' Law: $\frac{da}{dN} = C(\Delta K)^m$ where $K$ denotes the SIF. Accurate determination of the SIF for a given crack geometry and loading condition is therefore indispensable for predicting a component's remaining fatigue life, conducting damage tolerance assessments, and designing against fracture.

While analytical solutions for SIFs exist for simple geometries and loading conditions, real-world engineering structures often feature complex shapes, multiple cracks, and intricate boundary conditions. To handle such complexities, numerical methods, particularly the Finite Element Method (FEM), are widely used to compute SIFs (Dixon & Pook, 1969) with high accuracy. However, high-fidelity FEM simulations are often computationally expensive and time-consuming, making them impractical for large-scale structures or applications requiring rapid evaluation, such as design optimization or real-time structural health monitoring. Although experimental techniques can offer

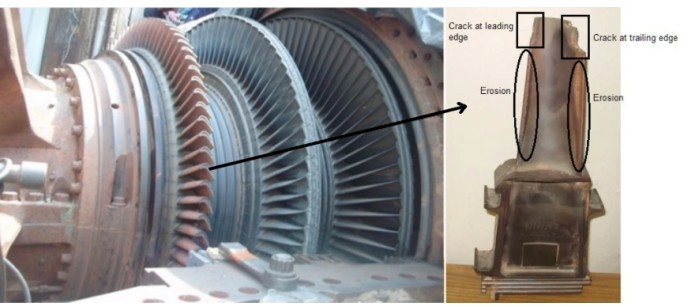

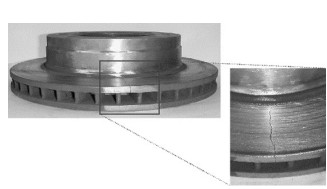

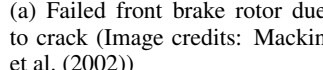

(a) Failed front brake rotor due to crack (Image credits: Mackin et al. (2002))

(b) Crack and erosion in a gas turbine blade (Image credit: Rani et al. (2017))

Figure 1: Cracks in brake rotor and turbine blade.

valuable insights, they are typically costly and limited in applicability due to constraints related to equipment, testing environments, and the range of accessible parameters.

As a result, engineers and researchers frequently rely on handbook solutions manually crafted by domain experts. These solutions serve as efficient surrogate models, offering reasonably accurate estimates of SIFs. For instance, the classical Raju-Newman (Newman Jr & Raju, 1981; Raju & Newman, 1979) and Fawaz & Andersson (2004) equations, expressed as polynomial functions, are capable of predicting SIFs for simple crack geometries such as semi-elliptical surface cracks and quarter-elliptical corner cracks near holes. Similarly, Pommier et al. (1999) provided a set of engineering formulas for Mode I surface cracks, while Wang & Lambert (1995) constructed local weight functions to compute SIFs for semi-elliptical surface cracks in finite-thickness plates. However, these handbook solutions are confined to simple geometries and loading conditions. These limitations highlight the pressing need for more flexible, general, and automated methods that can accommodate a broader range of geometric variations and complex loading scenarios — without requiring tedious and nontrivial manual derivations (Gautam et al., 2025).

Machine learning (ML) has recently emerged as a promising approach for SIF prediction. ML models, such as deep neural networks, offer great flexibility to handle complex geometries, boundary conditions, and loading scenarios through automated training, while also demonstrating improved accuracy (Zhang et al., 2023). Notable recent contributions include the work of Xia et al. (2022), who employed a convolutional neural network (CNN) to predict Mode I SIFs in coal rocks; and Xu et al. (2022), exploring Gaussian process (GP) regression (Williams & Rasmussen, 2006), tree-based models, and feed-forward neural networks for probabilistic failure risk assessment in aeroengine disks. Their hybrid model improved SIF prediction accuracy by 5%–35% compared to the widely adopted universal weight function (Glinka & Shen, 1991). In addition, Zhang et al. (2023) used a multi-layer feed-forward neural network to predict mixed-mode SIFs in composite materials.

However, ML approaches are data-driven and their success depends critically on the availability of rich, high-quality training data (Merrell et al., 2024). Given analytical solutions are limited and experimental data is scarce and costly to collect, the training datasets are mainly generated using high-fidelity FEM (Galić et al., 2018). However, generating accurate FEM data for SIFs is far from trivial (Liu et al., 2004). It demands domain expertise in meshing, particularly near crack tips where stress singularities arise. Accurately capturing the stress field in these regions often requires extremely fine meshes or specialized elements, e.g., quarter-point singular elements. The computation can be highly costly and time-consuming (Bathe, 2006), especially when dealing with complex 3D geometries or conducting parametric studies varying crack dimensions and locations. Furthermore, extracting SIFs from FEM results involves sophisticated post-processing techniques, e.g., (Courtin et al., 2005; Fu et al., 2012; Hou et al., 2022), which adds another layer of complexity and potential for error. Collectively, the high cost, required expertise, and time-consuming nature of this process create a substantial barrier to advancing ML methods in this domain.

To address these challenges, we introduce:

- **SIFBench** — an open-source, large-scale benchmark database designed to support the development and evaluation of ML methods for SIF prediction and fatigue analysis. The database contains an extensive collection of high-fidelity FEM-derived SIF solutions, cov-

ering approximately 5 million unique cracks, geometries, and loading conditions across 37 distinct scenarios representative of operational aircraft wing environments. Figure 1 shows cracks in critical components like brake rotors and turbine blades. The datasets we provide are representative of the cracks noticed commonly in the airframe or fuselages but are not too different from the scenarios shown in Figure 1.

- **A unified and convenient Python interface** — enabling users to easily access and customize the datasets tailored to specific use cases.
- **Baseline ML models** —— including random forests, support vector regression, feedforward neural networks, and Fourier neural operator — applied to the datasets to provide reference results and a solid starting point for further development.
- **Comprehensive evaluation metrics** — to facilitate thorough assessment of predictive performance across diverse models and tasks.
- **Source code templates and examples** — covering data loading, model training, validation, and performance evaluation to streamline downstream development.

By substantially lowering the entry barrier, SIFBench is poised to accelerate progress in ML-based fatigue analysis and unlock new applications across structural integrity assessment, predictive maintenance, and beyond.

## 2 SIFBENCH: A BENCHMARK FOR MACHINE LEARNING-BASED FATIGUE ANALYSIS

We now introduce SIFBench, a benchmark database developed to support the advancement and validation of ML methods for SIF prediction. The database contains approximately 5 million unique configurations of 3D cracks, geometries, and loading conditions, organized into 37 distinct scenarios — each defined by a specific combination of geometry type and loading setup. These scenarios are divided into single-crack and twin-crack cases, comprising roughly 1.6 million and 3.3 million geometries, respectively.

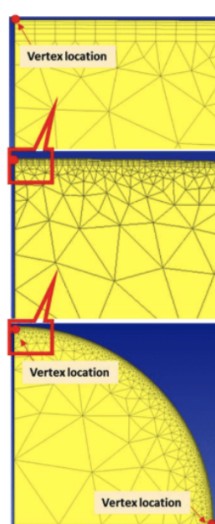

The single-crack category includes semi-elliptical surface cracks in a finite plate subjected to tensile loading, as well as various quarter-elliptical and through-thickness corner cracks located at straight and countersunk bolt holes. Most configurations are analyzed under three independent loading conditions: *tension*, *bending*, and *bearing*. The twin-crack category consists of paired quarter-elliptical and through-thickness corner cracks, also positioned at straight and countersunk bolt holes and evaluated under the same three loading conditions, each applied separately.

All the datasets (except the surface crack dataset) were obtained from the Center for Aircraft Structural Life Extension (CAStLE) at the U.S. Air Force Academy (Fawaz & Andersson, 2004). The meshes were specifically designed for the hp-version of the FEM (Babuška & Suri, 1990), allowing high accuracy in stress field resolution (Zeng & Wiberg, 1992). For the problems considered, the relative errors upon mesh convergence are consistently around 0.05%. Furthermore, the solvers were validated against cases with known analytical solutions, where the FEM solutions

Figure 2: Mesh pattern for the 2000 element edges along the crack front

achieved relative errors consistently below 0.03% (Litvinov et al., 2019). Fawaz et al. (2003) also verified the results experimentally using specimens made from 7075-T651 bare aluminum plate and found a good agreement between the simulations and the experiments. Figure 2 shows the FEM mesh, which was refined near the crack front to accurately capture the complex stress fields. To assess the accuracy of the dataset, several analyses were conducted at CAStLE and by other industry experts (Pilarczyk et al., 2022), to the extent that this dataset is now regarded as the most reliable simulation dataset available for SIFs (Pilarczyk et al., 2020). More details related to the dataset and the FEM simulations are shown in Appendix.

In the twin crack database, all the cases are 12 dimensional. For the single crack database, surface crack case is 5 dimensional and all the other cases are 6 dimensional. The scenarios increase in complexity in the following order: Surface Crack in a Rectangular Plate (Section 2.1.1) < Corner

Crack from a Straight Shank-Hole in a Plate (Section 2.1.2) < Corner Crack from a Countersunk Hole in a Plate (Section 2.1.3) < Twin-Crack Cases (Section 2.2). This progression in difficulty is primarily due to:

- **Geometric complexity**: Structural features such as holes—and especially countersunk holes—induce complex stress concentrations that demand more detailed modeling.
- **Loading conditions**: Compared to simple tension, non-uniform loading conditions (e.g., bending, bearing) lead to more varied and challenging stress distributions.
- **Number of cracks**: The presence of multiple cracks introduces crack interaction effects, which complicate the stress field and require more sophisticated analysis.

## 2.1 SINGLE-CRACK DATASETS

We first present the single-crack datasets, where each component contains one crack. These datasets were generated using high-fidelity FEM simulations, and span three distinct geometry types, varying in both component and crack shapes.

### 2.1.1 SURFACE CRACK IN A RECTANGULAR PLATE

The first type represents semi-elliptical surface cracks embedded in rectangular plates subjected to Mode I tensile loading Merrell et al. (2024). The SIFs ($K_I$) were extracted from FEM-computed displacement fields using Abaqus[1], in conjunction with the fracture mechanics software FRANC3D (Wawrzynek et al., 2010). The $K_I$ values were calculated from the stress fields via the method in (Hou et al., 2022).

This dataset contains 2,956 unique simulations, each defined by distinct plate and crack geometries as illustrated in Figure 3. In total, approximately

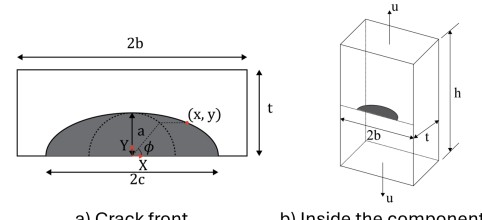

Figure 3: Surface crack in a rectangular plate.

100,000 SIF values were generated. Each crack geometry is characterized by three dimensionless parameters: $a/t$, $a/c$, and $c/b$, where $a$ is the crack depth, $c$ is the surface half-length, $t$ is the plate thickness, and $b$ is the plate width. The range of these parameters is summarized in Appendix Table 3. Individual points along the crack front are identified by the angle $\phi$, measured with respect to the inscribed circle.

### 2.1.2 CORNER CRACK FROM A STRAIGHT SHANK-HOLE IN A PLATE

The second geometry type involves corner cracks originating at a straight shank-hole in a plate. Three distinct loading conditions were considered independently: tension, bending, and bearing/pin loading. Figure 4 illustrates the plate and hole geometry under each of these loading scenarios. There are two crack configurations:

**Quarter-Elliptical Corner Crack.** In the first case, one end of the crack front is at the front of the plate and and the other end is at the hole, forming a quarter-elliptical shape (see Figure 5a). The dataset comprises 28,781 simulations, each representing a unique combination of plate and crack geometries. The geometries are described by the dimensionless

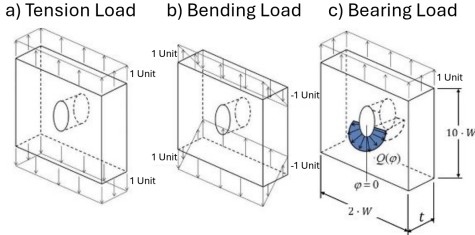

Figure 4: A load of unit 1 acts on the plate with a straight shank-hole.

parameters $W/R$, $a/c$, $a/t$, and $R/t$, where $W$ is the plate width, $R$ is the hole radius (fixed to 10 units), $a$ is the crack depth, $c$ is the surface half-length, and $t$ is the plate thickness. SIF values along the crack front are recorded using the parametric angle $\phi$, sampled at a resolution of 0.024 radians. Additional information is provided in Appendix Table 4.

---

[1] https://www.3ds.com/products/simulia/abaqus

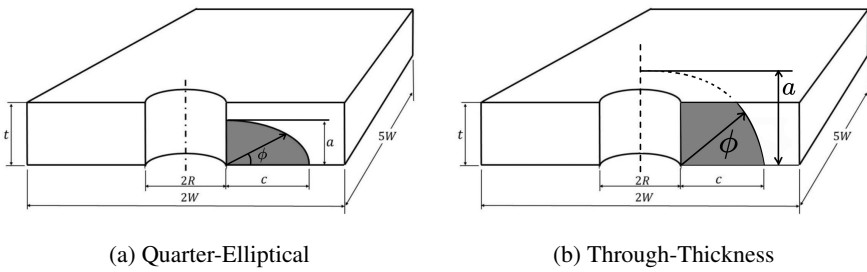

(a) Quarter-Elliptical          (b) Through-Thickness

Figure 5: Single corner crack from a straight shank-hole in a plate. Shaded is the crack front.

**Through-Thickness Corner Crack.** In the second case, the two ends of the crack front are at the front and the back of the plate (see Figure 5b). The dataset includes 5,426 distinct simulations, with geometries characterized by the same parameters: $W/R$, $a/c$, $a/t$, and $R/t$. As with the previous case, SIF values are provided along the crack front as a function of the parametric angle $\phi$, with a resolution of 0.024 radians. Additional information is listed in Appendix Table 5.

### 2.1.3 CORNER CRACK FROM A COUNTERSUNK HOLE IN A PLATE

The third geometry type involves corner cracks originating from a countersunk hole in a plate. As in Section 2.1.2, FEM results were obtained from the Center for Aircraft Structural Life Extension (CAStLE) at the U.S. Air Force Academy (Fawaz & Andersson, 2004), using hp-version finite element meshes tailored for high accuracy (Babuška & Suri, 1990). Tension, bending, and bearing/pin loading conditions were applied independently. The overall plate and hole geometry under these loading conditions is shown in Figure 6a. The datasets contain four types of countersunk holes, each defined by a different slant start depth. As shown in Figure 6b, this depth is captured by the non-dimensional parameter $b/t$, where $b$ is the vertical distance from the front of the plate to the start of the countersink, and $t$ is the total thickness of the plate. Two crack configurations are considered:

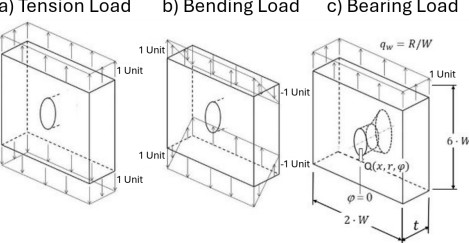

(a) A load of unit 1 acts on the plate with a countersunk shank-hole.

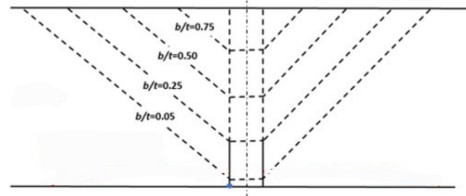

(b) Four types of countersunk holes.

Figure 6: A plate with a countersunk hole.

**Quarter-Elliptical Corner Crack.** In this configuration, the two ends of the crack front are at the front of the plate and at the cylindrical section of the hole, forming a quarter-elliptical shape. The geometry of this configuration is shown in Figure 7a. The finite element models generate 114,442; 127,884; 85,360; and 72,275 unique plate and crack geometries for $b/t = 0.75$, 0.5, 0.25, and 0.05, respectively. Each simulation is characterized by the following dimensionless parameters: $W/R$, $a/c$, $a/t$, $r/t$ where $W$ is the plate width, $R$ is the hole radius, $a$ is the crack depth, and $c$ is the surface length of the quarter-elliptical crack. SIFs are computed along the crack front and represented using the parametric angle $\phi$, which tracks the position along the crack front. The values are sampled at a resolution of 0.024 radians. Detailed statistics for each $b/t$ case are provided in Appendix Tables 6—9.

**Through-Thickness Corner Crack.** In the second configuration, the two ends of the crack front are at the front of the plate and the countersink or the back of the plate. The geometry is shown in Figure 7b. The finite element models generate 22,417 and 84,265 distinct plate and crack geometries for $b/t = 0.5$ and 0.05, respectively. The same geometric features as above — $W/r$, $a/c$, $a/t$, and $r/t$ — are used to describe the configurations. The corresponding SIF values are extracted along the crack front using the angle $\phi$ at a resolution of 0.024 radians. More details are provided in Appendix Tables 10 and 11.

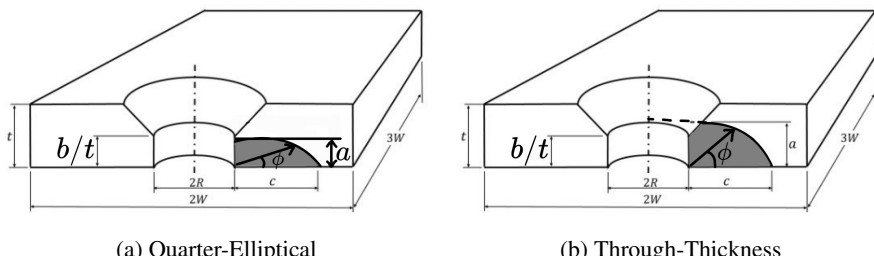

(a) Quarter-Elliptical     (b) Through-Thickness

Figure 7: Single corner crack from a countershank hole in a plate. Shaded is the crack front.

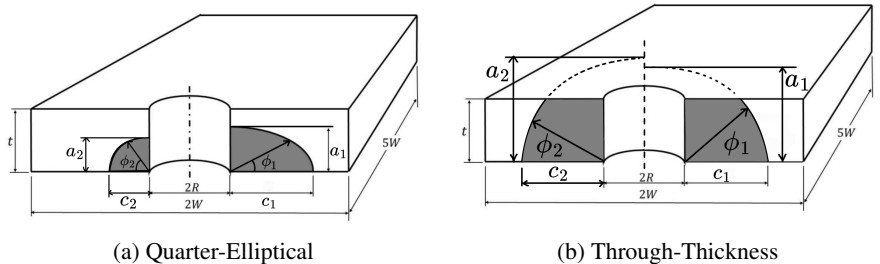

(a) Quarter-Elliptical     (b) Through-Thickness

Figure 8: Twin corner cracks from a straight shank-hole in a plate. Shaded are crack fronts.

## 2.2 TWIN-CRACK DATASETS

We next present the twin-crack datasets, in which each instance includes a pair of cracks. These datasets span two geometry types, as described in Sections 2.1.2 and 2.1.3, with two crack configurations considered for each type. The same FEM methods described earlier were used to generate all datasets.

**Quarter-Elliptical Corner Cracks from a Straight Shank Hole in a Plate.** The crack geometry is illustrated in Figure 8a. The FE models generate 72,000 unique plate and crack configurations, each represented by the ratio of plate width to hole radius ($W/r$), the aspect ratios of the two cracks ($a_1/c_1$ and $a_2/c_2$), the relative crack depths ($a_1/t$ and $a_2/t$), and the ratio of hole radius to plate thickness ($r/t$). SIF values are recorded along both crack fronts and are parameterized by the angles $\phi_1$ and $\phi_2$, each sampled at a resolution of 0.024 radians. Summary statistics for this dataset are provided in Appendix Table 12.

**Through-Thickness Corner Cracks from a Straight Shank Hole in a Plate.** The corresponding geometry is shown in Figure 8b. This dataset consists of 3,055 simulations, with each configuration described by the same set of features: $W/r$, $a_1/c_1$, $a_1/t$, $a_2/c_2$, $a_2/t$, and $r/t$. As before, SIF values are evaluated along the crack fronts using the parametric angles $\phi_1$ and $\phi_2$, with a resolution of 0.024 radians. Summary statistics are presented in Appendix Table 13.

**Quarter-Elliptical Corner Cracks from a Countersunk Hole in a Plate.** This configuration, shown in Figure 9a, includes 1,091,419 plate and crack geometries generated for $b/t = 0.5$. Each geometry is described by $W/r$, $a_1/c_1$, $a_1/t$, $a_2/c_2$, $a_2/t$, and $r/t$, with SIF values reported along the two crack fronts using $\phi_1$ and $\phi_2$ as the parametric angles (again sampled at 0.024 radians). Summary statistics for this dataset appear in Appendix Table 14.

**Through-Thickness Corner Cracks from a Countersunk Hole in a Plate.** The geometry for this case is shown in Figure 9b. The dataset contains 2,720 simulations for $b/t = 0.5$. Each case is described using the same set of geometric features—$W/r$, $a_1/c_1$, $a_1/t$, $a_2/c_2$, $a_2/t$, and $r/t$—and SIF values parameterized along the crack fronts using angles $\phi_1$ and $\phi_2$ with a resolution of 0.024 radians. Summary statistics are listed in Appendix Table 15.

## 2.3 OVERVIEW OF METRICS

We introduce three metrics to enable a comprehensive evaluation of SIF prediction. Let $y_\phi$ denote the FEM-computed SIF value at the angle $\phi$ (i.e.,ground-truth), and $\hat{y}_\phi$ denote the prediction of an ML method given the input, including geometric parameters, loading conditions, and $\phi$.

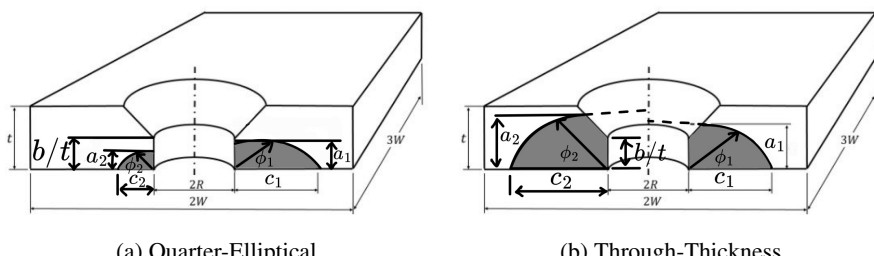

(a) Quarter-Elliptical          (b) Through-Thickness

Figure 9: Twin corner cracks from a countershank hole in a plate. Shaded are the crack fronts.

- **Normalized Absolute Error (NAE):** defined as $|y_\phi - \hat{y}_\phi|/|y_\phi|$.

- **Relative $L_2$ Error:** a metric used to evaluate the predictive performance on a whole crack. Denote all the ground-truth SIF values computed at the crack front by $\{y_{\phi_1}, \ldots, y_{\phi_N}\}$. The relative $L_2$ error is defined as: $\sqrt{\sum_{i=1}^N (y_{\phi_i} - \hat{y}_{\phi_i})^2}/\sqrt{\sum_{i=1}^N y_{\phi_i}^2}$.

- **Complementary Cumulative Density Function of NAE (CCDF-NAE):** A metric used to evaluate predictive performance across an entire dataset encompassing diverse crack geometries, component shapes, and loading conditions. It is defined as $1 - \text{CDF}(\epsilon)$ where the CDF is the cumulative density function of the model's NAE values across the dataset. For any error $\epsilon$, a smaller CCDF-NAE indicates better performance.

## 2.4 BASELINE METHODS

The baseline methods fall into two categories. The first class comprises models that predict a single SIF value at a time. Given geometry parameters, loading conditions, and a specific angle $\phi$ — together forming a feature vector — the model predicts the SIF value of the crack front corresponding to $\phi$. Methods in this class include **Random Forest Regression (RFR)**, **Support Vector Regression (SVR)**, and **Feedforward Neural Networks (FNN)**. The second class consists of recent neural operators (Azizzadenesheli et al., 2024), which learn mappings between functional spaces. Here, the SIF along the full crack front is treated as a function of angle $\phi$. The input includes the geometry parameters, loading conditions, and a set of angles $\phi_1, \ldots, \phi_N$ for which predictions are desired. The output is the corresponding set of $N$ predicted SIF values. We adopt the **Fourier Neural Operator (FNO)** (Hou et al., 2022) as a representative model from this class.

## 2.5 DATA FORMAT AND ACCESS

The benchmark database comprises a collection of CSV files (Mitlöhner et al., 2016), each corresponding to a distinct type of component-crack geometry. Each file includes multiple columns detailing the crack geometry, component geometry, loading, parametric angle ($\phi$), and the associated SIF values. All the datasets will be hosted on Hugging Face and readily accessible for download. A detailed README file would be included in the repository to guide users through the dataset structure and usage. The data can be conveniently accessed using the Python Pandas library[2], which also supports flexible customization, modification, and analysis. Listing 1 shows a code snippet that loads the surface crack dataset described in Section 2.1.1 and trains a Random Forest Regression (RFR) model on it. The provided scripts can be easily extended to other datasets and machine learning models. Comprehensive training and testing scripts will be made available in the dedicated GitHub repository, lowering the barrier for users to experiment with various datasets and modeling approaches.

Listing 1: Training RFR on surface crack dataset

```python
import pandas as pd
from sklearn.ensemble import RandomForestRegressor

df = pd.read_csv("SURFACE_CRACK_TRAIN.csv")
# Drop crack index
```

---

[2]https://pandas.pydata.org/

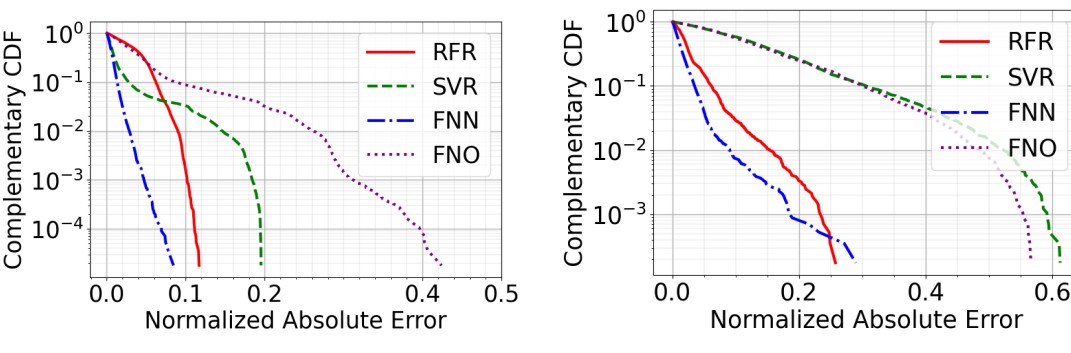

(a) SC under tension loading.     (b) TT CC (straight) under tension loading.

Figure 10: CCDF-NAE.

```
d = df.to_numpy()[:,1:]
# Train
rfr = RandomForestRegressor(max_depth=None)
rfr.fit(d[:,:-1], d[:,-1]))
# Test
df_test = pd.read_csv("SURFACE_CRACK_TEST.csv")
df_test = df_test.to_numpy()[:,1:]
y_pred = rfr.predict(d[:,:-1])
```

## 3 ML BENCHMARKS

This section presents a selection of experiments conducted on our SIFBench datasets, providing reference results and baselines for future development. Specifically, Tables 1 and 2 report the relative $L_2$ errors for the single-crack and twin-crack datasets, respectively. The mean NAE values are provided in Appendix Tables 16 and 17. For each case, we have trained and tested the models on a subset of the data with details in Appendix Section A.3.

In single-crack scenarios, the Feedforward Neural Network (FNN) often achieves the best performance among the tested methods, whereas in twin-crack datasets, Random Forest Regression (RFR) generally performs best. Although the neural operator model — Fourier Neural Operator (FNO) — is conceptually appealing due to its ability to model functional mappings, it frequently lags behind the more straightforward single-value prediction models in terms of accuracy. Figures 10a and 10b illustrate the Complementary Cumulative Density Function of NAE (CCDF-NAE) for each method on the surface cracks in a plate and the through-thickness corner cracks from a straight bolt hole in a plate under tension loading, respectively.

Overall, these preliminary results highlight the potential of machine learning methods for SIF prediction — for instance, many relative $L_2$ errors are within a few percent. However, for these methods to be viable in broader engineering and scientific applications, a more stable relative $L_2$ error on the order of $10^{-4}$ or lower is generally required. This underscores the need for continued improvement and the development of more powerful machine learning methodologies.

## 4 CONCLUSION

We have introduced SIFBench— a novel benchmark database designed to advance machine learning development and applications in stress intensity factor (SIF) prediction and fatigue analysis. The database includes over five million unique crack and component geometries spanning 37 distinct scenarios. It is accompanied by a user-friendly Python interface and sample training and evaluation code, significantly lowering the barrier to entry for both researchers and engineers. Preliminary experiments demonstrate the strong potential of ML methods in this direction. By providing standardized datasets, evaluation metrics, and template code, SIFBench is well-positioned to fos-

Table 1: Relative $L_2$ errors on selected single-crack datasets. Abbreviations: QE = Quarter-Elliptical, TT = Through-Thickness, CC = Corner Crack, SC = Surface Crack.

| Scenario | Loading | RFR | SVR | FNN | FNO |
|---|---|---|---|---|---|
| QE SC (Finite Plate) | Tension | 0.0355 | 0.0156 | 0.0076 | 0.0451 |
| QE CC (Straight Hole) | Tension | 0.0210 | 0.0847 | 0.0145 | 0.0680 |
| | Bending | 0.0529 | 0.2465 | 0.0616 | 0.1275 |
| | Bearing | 0.0364 | 0.2478 | 0.0362 | 0.2049 |
| TT CC (Straight Hole) | Tension | 0.0294 | 0.1601 | 0.0184 | 0.1550 |
| | Bending | 0.0268 | 0.9059 | 0.0719 | 0.1751 |
| | Bearing | 0.0674 | 0.5235 | 0.0889 | 0.2706 |
| QE CC (Countersunk Hole; $b/t = 0.75$) | Tension | 0.0195 | 0.1448 | 0.0138 | 0.2509 |
| | Bending | 0.0219 | 0.1250 | 0.0118 | 0.2037 |
| | Bearing | 0.0301 | 0.1592 | 0.0168 | 0.2229 |
| QE CC (Countersunk Hole; $b/t = 0.5$) | Tension | 0.0231 | 0.1798 | 0.0232 | 0.1795 |
| | Bending | 0.0256 | 0.2187 | 0.0146 | 0.2111 |
| | Bearing | 0.1771 | 0.5372 | 0.3090 | 0.4605 |
| TT CC (Countersunk Hole; $b/t = 0.5$) | Tension | 0.1671 | 0.1634 | 0.1757 | 0.1174 |
| | Bending | 0.2261 | 0.4616 | 0.3021 | 0.2079 |
| | Bearing | 0.1757 | 0.1758 | 0.1738 | 0.1175 |
| QE CC (Countersunk Hole; $b/t = 0.25$) | Tension | 0.0198 | 0.1595 | 0.0144 | 0.2377 |
| | Bending | 0.0240 | 0.1301 | 0.0139 | 0.2138 |
| | Bearing | 0.0305 | 0.1681 | 0.0173 | 0.2572 |
| QE CC (Countersunk Hole; $b/t = 0.05$) | Tension | 0.0179 | 0.1590 | 0.0142 | 0.2271 |
| | Bending | 0.0189 | 0.1366 | 0.0111 | 0.2219 |
| | Bearing | 0.0297 | 0.1498 | 0.0138 | 0.2226 |
| TT CC (Countersunk Hole; $b/t = 0.05$) | Tension | 0.0284 | 0.1229 | 0.0373 | 0.0888 |
| | Bending | 0.0286 | 0.1351 | 0.0416 | 0.1073 |
| | Bearing | 0.0352 | 0.1373 | 0.0377 | 0.0910 |

Table 2: Relative $L_2$ errors on selected twin-crack datasets. C1 indicates the error for crack at the right side of the hole, and C2 at left side. The slant start depth of the countersunk hole datasets is $b/t = 0.5$.

| Scenario | Loading | RFR | | SVR | | FNN | | FNO | |
|---|---|---|---|---|---|---|---|---|---|
| | | C1 | C2 | C1 | C2 | C1 | C2 | C1 | C2 |
| QE CC (Straight) | Tension | 0.020 | 0.020 | 0.314 | 0.320 | 0.036 | 0.031 | 0.261 | 0.264 |
| | Bending | 0.026 | 0.027 | 0.468 | 0.474 | 0.039 | 0.035 | 0.338 | 0.353 |
| | Bearing | 0.021 | 0.021 | 0.559 | 0.575 | 0.028 | 0.040 | 0.365 | 0.359 |
| TT CC (Straight) | Tension | 0.034 | 0.033 | 0.236 | 0.230 | 0.038 | 0.045 | 0.208 | 0.183 |
| | Bending | 0.032 | 0.031 | 0.915 | 0.930 | 0.203 | 0.076 | 0.216 | 0.221 |
| | Bearing | 0.039 | 0.043 | 0.591 | 0.589 | 0.038 | 0.041 | 0.326 | 0.332 |
| QE CC (Countersunk) | Tension | 0.013 | 0.013 | 0.080 | 0.079 | 0.014 | 0.014 | 0.060 | 0.056 |
| | Bending | 0.013 | 0.013 | 0.110 | 0.112 | 0.012 | 0.012 | 0.097 | 0.100 |
| | Bearing | 0.015 | 0.015 | 0.104 | 0.101 | 0.013 | 0.016 | 0.056 | 0.057 |
| TT CC (Countersunk) | Tension | 0.028 | 0.020 | 0.095 | 0.077 | 0.077 | 0.084 | 0.108 | 0.088 |
| | Bending | 0.043 | 0.029 | 0.159 | 0.171 | 0.035 | 0.033 | 0.138 | 0.139 |
| | Bearing | 0.033 | 0.026 | 0.092 | 0.078 | 0.074 | 0.303 | 0.076 | 0.081 |

ter innovation and accelerate the development of more accurate and efficient methods for structural integrity assessment and predictive maintenance in critical industries.

**Limitations and Future Work:** While SIFBench covers a wide range of scenarios, it does not yet fully capture the complexity of real-world component shapes and crack geometries, crack interactions beyond twin cracks, or a broader spectrum of loading conditions, such as dynamic, thermal, or multi-axial stresses. We plan to expand the database by conducting high-fidelity finite element simulations on these more complex cases and integrating real-world experimental data to further enhance its utility. As this work primarily focuses on providing benchmark datasets for SIF prediction, we have limited it to SIF values. However, in future work, we will work on releasing a well-organized dataset of full stress fields. Such datasets would be substantially larger and contain much richer information, enabling broader applications.

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

# A  APPENDIX

## A.1  BACKGROUND

The stress intensity factor (SIF) is a fundamental parameter in linear elastic fracture mechanics (LEFM), used to characterize the stress state at the tip of a crack in a linear elastic material (Perez, 2004). It quantifies the magnitude of the singular stress field near the crack tip, which governs crack propagation. LEFM assumes that plastic deformation is confined to a small region around the crack tip compared to the crack size and component dimensions (Anderson & Anderson, 2005). The SIF depends on the applied loading, crack size, and orientation, and the geometry of the component (Wang, 1996). There are three basic modes of crack face displacement — Mode I (opening mode): tensile stresses are applied perpendicular to the crack plane; Mode II (sliding mode): in-plane shear stresses act parallel to the crack plane and perpendicular to the crack front; Mode III (tearing mode): out-of-plane shear stresses act parallel to both the crack plane and crack front. The corresponding SIFs are denoted by $K_I$, $K_{II}$, and $K_{III}$, respectively.

The general expression for SIF is given by $K = Y\sigma\sqrt{\pi a}$ (Wang, 1996) where $\sigma$ is the nominal applied stress, $a$ is a characteristic dimension of the crack (e.g.,half the length of an internal crack or the full length of an edge crack), and $Y$ is a dimensionless correction factor that accounts for the geometry of the component and crack, and the type of loading. For example, in an infinite plate containing a central crack of length $2a$ subjected to a uniform tensile stress $\sigma$, the geometry factor is $Y = 1.0$, yielding the Mode I SIF: $K_I = \sigma\sqrt{\pi a}$. In more realistic, finite geometries or under non-uniform loading, $Y$ is often a more complex function, typically dependent on geometric ratios such as $a/W$, where $W$ is the width of the component. The critical stress intensity factor, known as the *fracture toughness* (denoted as $K_c$), is a material property that quantifies the material's resistance to fracture (Launey & Ritchie, 2009). Crack propagation is predicted to occur when the applied SIF reaches or exceeds the critical value, i.e.,$K \geq K_c$.

While analytical solutions or handbook values exist for many simple geometries and loading cases, calculating SIFs for complex, real-world components often requires numerical methods (Anderson & Anderson, 2005). The finite element method (FEM) is widely used for this purpose (Han et al., 2015). In FEM analysis, the component geometry, including the crack, is discretized into a mesh of finite elements, and the governing equations of elasticity are solved numerically to obtain the displacement and stress fields throughout the component under the applied loads and boundary conditions (Spencer, 2004). Special attention is paid to meshing around the crack tip, often using refined meshes or specialized singular elements to accurately capture the stress singularity. Once the near-tip stress and displacement fields are obtained, SIFs ($K_I$, $K_{II}$, $K_{III}$) can be extracted using various post-processing techniques. Standard methods include the displacement correlation method (Fu et al., 2012) — fitting the numerically obtained displacements near the crack tip to the theoretical LEFM displacement equations — and the J-integral method (Courtin et al., 2005) — evaluating the path-independent energy release rate integral $J$ around the crack tip, which is directly related to the SIFs under linear elastic conditions. Overall, FEM provides a powerful and flexible framework

for accurately computing SIFs in components with complex geometries and loading conditions, enabling reliable fracture assessments in engineering applications.

## A.2 DATASET DETAILS

The datasets were generated by the Center for Aircraft Structural Life Extension (CAStLE) at the U.S. Air Force Academy and have been developed over the course of more than 10 years. The simulations were performed on large internal CPU clusters and took over 5 million CPU hours in total (over the course of several years on multiple clusters in parallel). The Table 3-15 detail the geometric parameter space for thirteen distinct datasets characterizing various crack and plate configurations. The datasets cover surface cracks, single and twin quarter-ellipse corner cracks, and single and twin through-thickness corner cracks, considering both general straight bolt-hole (BH) and specific countersunk (CS) hole configurations defined by different $b/t$ ratios (0.75, 0.5, 0.25, 0.05). For each dataset, the tables specify the key dimensionless geometric features, such as crack aspect ratio ($a/c$), relative crack depth ($a/t$), relative crack length ($c/b$), width-to-radius ratio ($W/R$), and radius-to-thickness ratio ($R/t$). For twin crack configurations, parameters for both cracks (e.g., $a1/c1$, $a1/t$, $a2/c2$, $a2/t$) are defined. Each table outlines the minimum and maximum values explored for these features, establishing the bounds of the parameter space covered, along with the coarsest resolution or step size used between data points for each feature within that range. Collectively, these tables define the scope and granularity of the geometric variations included in each specific crack dataset.

Table 3: Feature information for the surface cracks in a rectangular plate.

| Feature | Min value | Max value | Resolution (coarsest) |
|---------|-----------|-----------|-----------------------|
| $a/c$ | 0.2 | 2 | 0.05 |
| $a/t$ | 0.2 | 0.85 | 0.05 |
| $c/b$ | 0.01 | 0.3 | 0.1 |

Table 4: Feature information for the single quarter-ellipse corner cracks from a straight shank-hole in a plate.

| Feature | Min value | Max value | Resolution (coarsest) |
|---------|-----------|-----------|-----------------------|
| $W/R$ | 1.6 | 1000 | 200 |
| $a/c$ | 0.1 | 10 | 1 |
| $a/t$ | 0.1 | 0.95 | 0.1 |
| $R/t$ | 0.1 | 10 | 2 |

Table 5: Feature information for the single through-thickness corner cracks from a straight shank-hole in a plate.

| Feature | Min value | Max value | Resolution (coarsest) |
|---------|-----------|-----------|-----------------------|
| $W/R$ | 10 | 1000 | 200 |
| $a/c$ | 0.1 | 10 | 1 |
| $a/t$ | 1,05 | 10 | 2 |
| $R/t$ | 0.1 | 10 | 2 |

Table 6: Feature information for single quarter-ellipse corner cracks from a countersunk hole in a plate, with $b/t = 0.75$.

| Feature | Min value | Max value | Resolution (coarsest) |
|---------|-----------|-----------|-----------------------|
| $W/R$ | 1.6 | 100 | 60 |
| $a/c$ | 0.1 | 10 | 4 |
| $a/t$ | 0.01 | 0.1 | 0.01 |
| $R/t$ | 0.1 | 9 | 1 |

Table 7: Feature information for single quarter-ellipse corner cracks from a countersunk hole in a plate, with $b/t = 0.5$.

| Feature | Min value | Max value | Resolution (coarsest) |
|---|---|---|---|
| $W/R$ | 1.6 | 100 | 60 |
| $a/c$ | 0.1 | 10 | 4 |
| $a/t$ | 0.01 | 0.5 | 0.1 |
| $R/t$ | 0.1 | 10 | 1 |

Table 8: Feature information for single quarter-ellipse corner cracks from a countersunk hole in a plate, with $b/t = 0.25$.

| Feature | Min value | Max value | Resolution (coarsest) |
|---|---|---|---|
| $W/R$ | 1.6 | 100 | 60 |
| $a/c$ | 0.1 | 10 | 4 |
| $a/t$ | 0.01 | 0.1 | 0.01 |
| $R/t$ | 0.1 | 10 | 1 |

Table 9: Feature information for single quarter-ellipse corner cracks from a countersunk hole in a plate, $b/t = 0.05$.

| Feature | Min value | Max value | Resolution (coarsest) |
|---|---|---|---|
| $W/R$ | 1.6 | 100 | 60 |
| $a/c$ | 0.1 | 10 | 4 |
| $a/t$ | 0.01 | 0.05 | 0.01 |
| $R/t$ | 0.1 | 10 | 1 |

Table 10: Feature information for single through-thickness corner cracks from a countersunk hole in a plate, with $b/t = 0.5$.

| Feature | Min value | Max value | Resolution (coarsest) |
|---|---|---|---|
| $W/R$ | 2.4 | 100 | 60 |
| $a/c$ | 0.1 | 10 | 4 |
| $a/t$ | 0.6 | 15 | 5 |
| $R/t$ | 0.2 | 5 | 2 |

Table 11: Feature information for single through-thickness corner cracks from a countersunk hole in a plate, with $b/t = 0.05$.

| Feature | Min value | Max value | Resolution (coarsest) |
|---|---|---|---|
| $W/R$ | 1.6 | 1605.5 | 800 |
| $a/c$ | 0.1 | 10 | 4 |
| $a/t$ | 0.06 | 0.95 | 0.01 |
| $R/t$ | 0.1 | 10 | 1 |

Table 12: Feature information of the twin quarter-ellipse corner cracks from a straight shank hole in a plate.

| Feature | Min value | Max value | Resolution (coarsest) |
|---|---|---|---|
| $W/R$ | 33.3 | 500 | 300 |
| $a1/c1$ | 0.1 | 10 | 2 |
| $a1/t$ | 0.1 | 0.95 | 0.1 |
| $a2/c2$ | 0.1 | 10 | 2 |
| $a2/t$ | 0.1 | 0.95 | 0.1 |
| $R/t$ | 0.2 | 3 | 1 |

Table 13: Feature information of the twin through-thickness corner cracks from a straight shank hole in a plate.

| Feature | Min value | Max value | Resolution (coarsest) |
|---|---|---|---|
| $W/R$ | 33.3 | 500 | 300 |
| $a1/c1$ | 0.1 | 10 | 2 |
| $a1/t$ | 1.05 | 5 | 1 |
| $a2/c2$ | 0.1 | 10 | 2 |
| $a2/t$ | 1.05 | 5 | 1 |
| $R/t$ | 0.2 | 3 | 1 |

Table 14: Feature information of the twin quarter-ellipse corner cracks from a countersunk hole in a plate, with $b/t = 0.5$.

| Feature | Min value | Max value | Resolution (coarsest) |
|---|---|---|---|
| $W/R$ | 33.3 | 500 | 300 |
| $a1/c1$ | 0.1 | 10 | 2 |
| $a1/t$ | 0.1 | 0.95 | 0.1 |
| $a2/c2$ | 0.1 | 10 | 2 |
| $a2/t$ | 0.1 | 0.95 | 0.1 |
| $R/t$ | 0.2 | 3 | 1 |

Table 15: Feature information of the twin through-thickness corner cracks from a countersunk hole in a plate, with $b/t = 0.5$.

| Feature | Min value | Max value | Resolution (coarsest) |
|---|---|---|---|
| $W/R$ | 2.4 | 100 | 60 |
| $a1/c1$ | 0.1 | 10 | 4 |
| $a1/t$ | 0.1 | 0.95 | 0.1 |
| $a2/c2$ | 0.1 | 10 | 4 |
| $a2/t$ | 0.6 | 0.95 | 0.1 |
| $R/t$ | 0.5 | 5 | 2 |

### A.3 ML Training Details

Tables 16 and 17 show the mean NAE for all the cases involving single and twin cracks, respectively. In each case the dataset was split into train-test with 75%-25% ratio, and training details for each ML algorithm are as follows:

1. **RFR**[3]**:** RFR model was trained using a subset of the full dataset. Specifically, a random sample of up to 100,000 data points was selected without replacement to form the training set. The input features for the model consisted of all columns representing the crack and plate geometries, while the target variable was SIF. During instantiation, the RFR model was configured with the *max_depth* hyperparameter set to *None*, allowing the individual decision trees within the forest to expand until all leaves were pure or contained fewer than the minimum samples required for a split. All other hyperparameters for the RFR, such as the number of trees (*n_estimators*) and criteria for splitting nodes, were kept at their default values as defined by the scikit-learn library. The model was then trained using the fit method on the prepared subset and subsequently saved for later use.

2. **SVR**[4]**:** SVR model training utilized a randomly selected subset of the data, also capped at a maximum size of 100,000 samples chosen without replacement from the complete dataset. The input features and the target output variable were defined identically to the RFR setup. The SVR model was instantiated using the *SVR()* constructor without any specified arguments, meaning all hyperparameters were set to their default values provided

---

[3]Trained using Ryzen 7 7700 CPU with 32 GB RAM
[4]Trained using Ryzen 7 7700 CPU with 32 GB RAM

by the scikit-learn library. These defaults typically include using the Radial Basis Function (rbf) kernel, a regularization parameter C of 1.0, and an epsilon value of 0.1, among others. Following initialization, the SVR model was trained on its respective data subset via the fit method and the resulting trained model was saved.

3. **NN**[5]**:** NN used was a deep, fully connected feedforward architecture designed for regression tasks. Its configuration was governed by two primary hyperparameters: the number of input features and the number of neurons in each hidden layer. The network consisted of five hidden layers with 15 neurons each, utilizing the leaky rectified linear unit (leaky ReLU) activation function. This particular activation function is chosen to address the vanishing gradient problem and to ensure that neurons can continue to learn even when not strongly activated. The model outputs a single SIF value. For training, the model employs a mean-squared-error loss function, which quantifies the difference between the predicted and actual values. Optimization is performed using Adam, which iteratively updates the network's parameters to minimize the loss. The training process is conducted over a predefined 150,000 epochs, with the model's performance assessed after each epoch on both the training and validation datasets. Progress is monitored and reported at regular intervals to provide feedback on the training and validation errors. An important aspect of the training procedure is the implementation of model checkpointing and early stopping. Whenever the validation loss improves, the current state of the model is saved, ensuring that the best-performing version is preserved. If the validation loss does not improve for a large number of consecutive epochs, an early stopping criterion is triggered to halt training, thereby preventing overfitting and saving computational resources. Throughout training, the model alternates between training and evaluation modes as appropriate, and gradient computation is disabled during validation to enhance efficiency. The training and validation losses are recorded at each checkpoint, enabling a detailed analysis of the model's learning dynamics. Collectively, these hyperparameters and training strategies are designed to optimize both the effectiveness and efficiency of the learning process, while safeguarding against overfitting and ensuring reproducibility.

4. **FNO**[6] The FNO model architecture is composed of several key hyperparameters: the number of input features (decided by the number crack and plate geometric features in the select dataset), the number of Fourier modes retained in the spectral convolution layers (we used 64), and the width of the network (we have used 64), which determines the number of channels in the lifted representation. The network begins by projecting the input data into a higher-dimensional feature space, followed by four consecutive layers that each combine a spectral convolution in the Fourier domain with a standard pointwise convolution. The spectral convolutions operate by transforming the input into the frequency domain, applying learnable complex-valued weights to a fixed number of Fourier modes, and then transforming the result back to physical space. This approach enables the model to efficiently capture both global and local patterns in the data. Each layer is followed by a non-linear activation function (GELU), and the final output is produced through two fully connected layers that map the features back to the desired output dimension. For training, the code utilizes a mini-batch approach with data loaded via PyTorch's DataLoader, and optimization is performed using the Adam algorithm with weight decay for regularization. The learning rate is scheduled to decay at fixed intervals, controlled by the step size and decay factor hyperparameters. The loss function employed is a relative $L^p$ loss, which measures the normalized difference between predicted and true outputs, providing a scale-invariant metric that is particularly suitable for function regression tasks. Training progress is monitored using the normalized root mean squared error (NRMSE) and normalized mean squared error (NMSE) on both training and test sets, with these metrics computed and stored after each epoch. The model incorporates checkpointing, saving the parameters whenever a new best test NRMSE is achieved, thus ensuring that the best-performing model is preserved. Throughout training, the network alternates between training and evaluation modes as appropriate, and gradient computations are disabled during validation phases to improve efficiency. The combination of spectral and pointwise convolutions, advanced loss metrics, learning rate scheduling, and robust checkpointing collectively enable the model to achieve high accuracy and generalization in learning complex mappings between functions.

---

[5]Trained using NVIDIA RTX3090 GPU
[6]Trained using NVIDIA RTX3090 GPU

Table 16: Mean NAE on selected single-crack datasets. Abbreviations: QE = Quarter-Elliptical, TT = Through-Thickness, CC = Corner Crack, SC = Surface Crack.

| Scenario | Loading | RFR | SVR | FNN | FNO |
|---|---|---|---|---|---|
| QE SC (Finite Plate) | Tension | 0.0341 | 0.0138 | 0.0063 | 0.0429 |
| QE CC (Straight Hole) | Tension | 0.0183 | 0.0783 | 0.0126 | 0.0654 |
| | Bending | 0.2494 | 1.7469 | 0.1756 | 0.5131 |
| | Bearing | 0.0578 | 0.8123 | 0.0718 | 0.7145 |
| TT CC (Straight Hole) | Tension | 0.0277 | 0.1458 | 0.0155 | 0.1437 |
| | Bending | 0.0420 | 2.0930 | 0.3294 | 0.7150 |
| | Bearing | 0.0610 | 0.5495 | 0.0800 | 0.2668 |
| QE CC (Countersunk Hole; $b/t = 0.75$) | Tension | 0.0168 | 0.1415 | 0.0124 | 0.2454 |
| | Bending | 0.0192 | 0.1159 | 0.0103 | 0.2020 |
| | Bearing | 0.0263 | 0.1694 | 0.0155 | 0.2127 |
| QE CC (Countersunk Hole; $b/t = 0.5$) | Tension | 0.0299 | 0.2242 | 0.0233 | 0.1907 |
| | Bending | 0.0271 | 0.2888 | 0.0189 | 0.2522 |
| | Bearing | 0.6051 | 1.4120 | 0.8022 | 0.7981 |
| TT CC (Countersunk Hole; $b/t = 0.5$) | Tension | 25.8989 | 214.8582 | 63.1247 | 102.6938 |
| | Bending | 11.7549 | 284.9972 | 107.7207 | 118.3158 |
| | Bearing | 13.3025 | 204.2497 | 44.5282 | 103.0246 |
| QE CC (Countersunk Hole; $b/t = 0.25$) | Tension | 0.0212 | 0.1895 | 0.0152 | 0.2408 |
| | Bending | 0.0211 | 0.1208 | 0.0122 | 0.2123 |
| | Bearing | 0.0334 | 0.2833 | 0.0189 | 0.2609 |
| QE CC (Countersunk Hole; $b/t = 0.05$) | Tension | 0.0155 | 0.1499 | 0.0123 | 0.2220 |
| | Bending | 0.0167 | 0.1261 | 0.0098 | 0.2199 |
| | Bearing | 0.0263 | 0.1464 | 0.0123 | 0.2165 |
| TT CC (Countersunk Hole; $b/t = 0.05$) | Tension | 0.0269 | 0.1220 | 0.0374 | 0.0888 |
| | Bending | 0.0270 | 0.2550 | 0.0386 | 0.2342 |
| | Bearing | 0.0366 | 0.1517 | 0.0431 | 0.1464 |

Table 17: Mean NAE on selected twin-crack datasets. C1 indicates the error for crack at the right side of the hole, and C2 at left side. The slant start depth of the countersunk hole datasets is $b/t = 0.5$.

| Scenario | Loading | RFR | | SVR | | FNN | | FNO | |
|---|---|---|---|---|---|---|---|---|---|
| | | C1 | C2 | C1 | C2 | C1 | C2 | C1 | C2 |
| QE CC (Straight) | Tension | 0.017 | 0.017 | 0.322 | 0.331 | 0.032 | 0.028 | 0.269 | 0.274 |
| | Bending | 0.073 | 0.134 | 2.263 | 3.181 | 0.117 | 0.149 | 1.890 | 1.966 |
| | Bearing | 0.017 | 0.018 | 0.932 | 0.984 | 0.029 | 0.040 | 0.449 | 0.460 |
| TT CC (Straight) | Tension | 0.032 | 0.031 | 0.223 | 0.217 | 0.033 | 0.039 | 0.196 | 0.173 |
| | Bending | 0.054 | 0.044 | 2.337 | 2.179 | 0.637 | 0.264 | 1.118 | 0.841 |
| | Bearing | 0.035 | 0.039 | 0.641 | 0.630 | 0.032 | 0.037 | 0.326 | 0.334 |
| QE CC (Countersunk) | Tension | 0.010 | 0.011 | 0.073 | 0.071 | 0.012 | 0.012 | 0.059 | 0.055 |
| | Bending | 0.013 | 0.015 | 0.170 | 0.190 | 0.015 | 0.015 | 0.297 | 0.407 |
| | Bearing | 0.012 | 0.012 | 0.107 | 0.102 | 0.013 | 0.015 | 0.060 | 0.058 |
| TT CC (Countersunk) | Tension | 0.025 | 0.017 | 0.085 | 0.070 | 0.069 | 0.076 | 0.100 | 0.082 |
| | Bending | 0.169 | 0.149 | 0.889 | 1.187 | 0.128 | 0.145 | 0.898 | 1.141 |
| | Bearing | 0.029 | 0.022 | 0.083 | 0.074 | 0.071 | 0.339 | 0.068 | 0.084 |

