# OpenReview forum: "SIFBench: An Extensive Benchmark for 3D Fatigue Analysis"
_ICLR.cc/2026/Conference — Submitted to ICLR 2026_

### Official Review · Reviewer_CxHo · 2025-10-31

**Soundness:** 3
**Presentation:** 3
**Contribution:** 2
**Rating:** 4
**Confidence:** 4

**Summary:**

This paper introduces a dataset that is challenging for standard ML models. Fatigue analysis is an important area of study where machine learning could have a large impact. The benchmark results indicate that ML predictors perform nearly as well on twin-crack problems as single-crack problems; even though it is much harder to simulate twin-crack systems. The benefits to the materials degradation research community are clear, but the impact of this paper in the ICLR community is not clear. More details analyzing the results of the benchmark models would provide insight as to what advances in machine learning could provide more accurate results.

**Strengths:**

This is a well thought out dataset for fatigue analysis with an original contribution of a novel dataset. The paper is clearly written with high amounts of detail in how the data was created; yet it is lacking in other details as mentioned below. The benchmark results on 3 standard models show that L2 error would need to be improved from the current level of 10^-1 or 10^-2; to the practically useful level of 10^-4. Evaluation metrics are provided.

Note on my rating: I think the paper is sound, but this is not the right audience. "I would not mind if the paper is accepted" is true, but I worry that it will have less impact at ICLR than it would in a materials science venue.

**Weaknesses:**

For an ICLR paper, there needs to be more motivation for the impact that this dataset has for the ML community (it’s clear that this is useful for the materials degradation community, so it seems that is where the paper should be published to get the right audience). As an ML researcher, after reading this paper, I am unlikely to take the time to learn more about the data in Huggingface - the paper hasn’t given me enough information for me to know whether the methods that I work with are potentially a good fit for this dataset. Sections 2.4, 2.5, and 3 should be the majority of the paper to detail what is challenging about the dataset for existing ML approaches. In addition to the L2 errors summarized in Table 1 and 2, it would be helpful to see examples of what the ML models are predicting well and what they are not predicting well. An analysis of the results would lead to more insight that would lower the bar to further research.

From the abstract, it would be helpful to have some hints about what is challenging for ML in this application? Why does the ML community want to consider this dataset? Instead of “We report baseline results…”, tell us what you conclude from those results.

Data format section could use more details. How many features (paper says “multiple columns”)? Are the features scalar, vector, binary? I am not sure what form the “component geometry” (for example) would take.

**Questions:**

Why are RFR and SVR trained on a subset of data? 100,000 data points out of a total of 5 million is a very small subset. Yet, it appears that RFR is out-performing both SVR and FNO, while quite competitive with FNN (it would be helpful to highlight the lowest error in the tables).

For Tables 1 and 2, is each model trained on one Scenario at a time, or is each model trained on all of the Scenarios combined together?

The twin-crack problems may be harder to simulate, but it seems that they are not significantly harder for the ML predictors. Is this correct?

The number of simulations per Scenario can vary from 2,000 to over 1,000,000. Does the number of simulations in the dataset correlate with the accuracy (or negatively correlate with the error) for that scenario?

---

> ### Author Response · Authors · 2025-11-24
>
> We thank the reviewer for their comments.
>
> > C1: Why are RFR and SVR trained on a subset of data? 100,000 data points out of a total of 5 million is a very small subset. Yet, it appears that RFR is out-performing both SVR and FNO, while quite competitive with FNN (it would be helpful to highlight the lowest error in the tables).
>
> R1: RFR and SVR were trained on a subset of 100,000 points primarily for computational efficiency and feasibility, as training on the full 5 million data points would require substantial resources and time, especially for SVR which scales poorly with dataset size. Despite the smaller subset, RFR’s strong performance relative to SVR, FNN, and FNO highlights its robustness and suitability for this regression task. We will update tables to highlight the lowest errors for clearer comparison.
>
> > C2: For Tables 1 and 2, is each model trained on one Scenario at a time, or is each model trained on all of the Scenarios combined together?
>
> R2: Each model reported in Tables 1 and 2 was trained and evaluated separately on each Scenario individually, not on the combined data from all Scenarios. This design allows assessment of model performance specific to each scenario’s geometric and loading conditions.
>
> > C3: The twin-crack problems may be harder to simulate, but it seems that they are not significantly harder for the ML predictors. Is this correct? The number of simulations per Scenario can vary from 2,000 to over 1,000,000. Does the number of simulations in the dataset correlate with the accuracy (or negatively correlate with the error) for that scenario?
>
> R3: We have not conducted a detailed analysis of these specific aspects in this initial work, as our primary goal is to provide a comprehensive dataset that the research community can utilize and build upon. Regarding twin-crack problems, while they may be more challenging to simulate, our preliminary findings indicate that ML predictors do not find them significantly harder to handle. Similarly, although the number of simulations per scenario varies widely, we have not yet systematically studied its correlation with prediction accuracy or error, leaving these important analyses open for future work.

---

> > ### Comment · Reviewer_CxHo · 2025-11-24
> > **Analysis needed so that ICLR community will know what research directions to explore with this dataset**
> >
> > The authors’ responses to my review and the other reviews reinforce my opinion that this paper is lacking key analysis that would make it useful for the ICLR community. Examples:
> >
> > RFR was trained on a subset of 100,000 out of 5 million data points for “computational efficiency and feasibility”; yet it shows strong performance. This leaves the obvious next question of whether the model run on more data would solve the problem. Yet the authors have not indicated that they will do this computation to answer the question.
> >
> > “We have not conducted a detailed analysis of these specific aspects in this initial work, as our primary goal is to provide a comprehensive dataset that the research community can utilize and build upon.” If the detailed analysis existed, then there would be a foundation for the research community to build on. With just a dataset, but no analysis, there is little clue about what direction another research team should take.
> >
> > Other reviewers asked about using baselines like PINN. Authors’ response is “GNNs, GraphMeshNets, PINNs, and related architectures are not appropriate baselines for this work”. **This is a serious issue with the paper: if the most obvious direction to take the research is already dismissed by the paper writers, then the ICLR community does not know what to do with this dataset, even after reading the paper.**
> >
> > Authors’ responses indicate that they are only making minor editing changes to the paper and no new analysis will be added.

---

### Official Review · Reviewer_11k7 · 2025-11-01

**Soundness:** 2
**Presentation:** 2
**Contribution:** 3
**Rating:** 4
**Confidence:** 5

**Summary:**

The paper introduces a benchmark database SIFBench designed to support ML-based SIF prediction for structures with representative geometries and loading conditions, which are obtained from finite element analysis. Python script interfaces are provided for data access and customization. Baseline results using RFs, SVM, FNN, and FNO are demonstrated. The work offers a standardized and scalable resource, and fosters the development and application of ML methods in damage tolerance design and predictive maintenance.
However, the geometries and loading conditions of the models are not explained in detail, and the comparison with the traditional handbook approach was not made. Given the rich space of geometries and loading conditions in practice, the power of the current approach over the empirical approach is not clear.

**Strengths:**

1 A comprehensive SIF database with 5 million 3D crack and component geometries derived from high-fidelity finite element simulations across 37 distinct scenarios are reported, laying the ground for exploration of ML methods applied in fatigue crack predictions.
2 A unified format is reported, organizing the information in SIFs.
3 Interface scripts and demonstration of ML uses are provided, lower the barrier of learning and understanding.

**Weaknesses:**

1 Although the authors mentioned that the dataset is open source. I did not find the database from the paper and in the link reported in an earlier arXiv paper with the same title (https://huggingface.co/datasets/tgautam03/SIFBench - 404 error is reported). Consequently, I cannot check the content and scripts for more information.
2 The advantage of the current approach over the traditional handbook one is not demonstrated. Given the complexities in the geometries and loading conditions, a better explanation of how the dataset serves as practical sources should be given.

**Questions:**

1 In practical setups, the space of geometries and loading conditions is very large. Does the current approach cover more space in design than the handbook approach, and is there improvement in accuracy (if yes, is that significant enough?).
2 In reality, LEFM does not precisely work, and will result in physical bias even using high-fidelity finite element modeling. Is the potential improvement in accuracy beyond the handbook solutions significant enough to overcome this bias?

---

> ### Author Response · Authors · 2025-11-24
>
> We thank the reviewer for their comments.
>
> > C1: Although the authors mentioned that the dataset is open source. I did not find the database from the paper and in the link reported in an earlier arXiv paper with the same title (https://huggingface.co/datasets/tgautam03/SIFBench - 404 error is reported). Consequently, I cannot check the content and scripts for more information.
>
> R1: Keeping ICLR guidelines in mind, we made the dataset repository private during the review process to maintain anonymity and compliance with the conference's double-blind review policies. Consequently, the link was removed to avoid premature public access. We plan to make the repository publicly accessible upon acceptance to ensure full transparency and reproducibility.
>
> > C2: The advantage of the current approach over the traditional handbook one is not demonstrated. Given the complexities in the geometries and loading conditions, a better explanation of how the dataset serves as practical sources should be given.
>
> R2: As mentioned in the introduction, handbook solutions are limited to relatively simple geometries and loading conditions, and do not exist for most of the complex geometries and scenarios provided in this study. Our dataset is derived from high-fidelity FE simulations validated extensively by domain experts, ensuring its accuracy and practical relevance. By covering over 5 million crack and component configurations across 37 scenarios, the dataset provides a valuable resource that fills critical gaps where traditional handbook approaches fall short, enabling machine learning methods to tackle complex fatigue analysis problems encountered in real engineering applications.
>
> > C3: In practical setups, the space of geometries and loading conditions is very large. Does the current approach cover more space in design than the handbook approach, and is there improvement in accuracy (if yes, is that significant enough?). In reality, LEFM does not precisely work, and will result in physical bias even using high-fidelity finite element modeling. Is the potential improvement in accuracy beyond the handbook solutions significant enough to overcome this bias?
>
> R3: Yes, our dataset covers a much larger design space than traditional handbook approaches. For simpler cases, such as surface and corner cracks at a straight hole in a plate, we achieved an order of magnitude improvement in accuracy (please see the reference [1]). Given the vast and complex space of geometries and loading conditions encountered in practice, this significant accuracy gain and broader coverage highlight the practical advantage of our approach over handbook solutions, enabling more reliable and generalizable fatigue analysis.
>
> [1] Tushar Gautam, Jacob Hochhalter, Shandian Zhe, Eric Lindgren, and Robert M Kirby. Developing
> robust stress intensity factor models using fourier-based data analysis to guide machine learning
> method selection and training. Engineering Fracture Mechanics, pp. 111387, 2025.

---

### Official Review · Reviewer_V1ev · 2025-11-01

**Soundness:** 3
**Presentation:** 3
**Contribution:** 3
**Rating:** 6
**Confidence:** 3

**Summary:**

This paper introduces SIFBench, a large-scale, open-source benchmark for stress intensity factor (SIF) prediction in 3D fatigue crack scenarios. It consists of over 5 million high-fidelity FEM simulations spanning 37 geometry/loading cases relevant to structural components (e.g., aircraft plates with surface and corner cracks). The benchmark includes a Python API, baseline results using common ML models, and several evaluation metrics. The goal is to enable scalable and standardized ML-based SIF prediction to support fatigue life assessment. The paper is clearly written, well-organized, and includes helpful figures and explanations. The presentation of datasets, metrics, and baseline models is generally clear.

**Strengths:**

High-quality dataset: Targets an important problem in fatigue analysis, with relevance to aerospace, civil, and mechanical engineering. Paper includes 5 million FEM simulations across realistic engineering configurations, a scale not seen in prior 3D SIF benchmarks.

Open-source and reproducible: Promises data/code availability with unified API and baseline templates, which is an important step toward community standardization.

Strong experimental foundation: Simulations come from validated CAStLE datasets, using hp-FEM with verified accuracy.

**Weaknesses:**

The limited diversity of benchmarks presents a significant challenge. Current scenarios are restricted to flat plate geometries with surface or corner cracks, failing to represent crucial factors in fatigue such as curved shells, anisotropic materials, plasticity effects, or dynamic/multi-axial loading. It would be helpful to explain the choices and its impact on real-world fatigue situations.

The SIFBench authors do not demonstrate that models trained on their data would perform well on real cracked components (e.g., no case study of applying a trained model to a known lab experiment). This raises questions of real-world relevance: the benchmark might optimize algorithms for its specific simulation settings, but those models could face a reality gap when confronted with experimental noise or unmodeled physics.

Only standard regressors and one FNO are tested. No graph-based, physics-informed, hybrid, or transformer-style models. No hyperparameter tuning or ablation studies. The evaluation doesn't explore why simpler models outperform FNO.

The analysis lacks generalization, as all evaluations rely on train/test splits within a single scenario. Consequently, there's no assessment of cross-scenario generalization, such as training with straight-hole cracks and testing with countersunk-hole cracks. This significantly weakens claims of broad utility.


The simulation pipeline remains proprietary, despite the open availability of data and code. The generation scripts (ABAQUS/FRANC3D setup) are not accessible. Consequently, reproducing or extending the dataset necessitates considerable domain expertise.

**Questions:**

Could the authors add cross-scenario generalization results (e.g., train on some geometry types and test on held-out ones)? This would better assess real-world applicability.

Why were Transolver/transolver++ or other recent physics-based models or graph networks or neural operators not included in the baseline?


Report ML model’s memory usage, latency, inference throughput, and model size, wich are essential for practical deployment. SIFBench reports no such information, making it hard to compare models on real-world viability or cost-performance tradeoffs.

Will there be lightweight subsets or APIs to query the 5M dataset for users with limited compute?

How can the community extend the benchmark (e.g., contribute new simulations or experimental data)?


Efficiency Profiling: Can the authors report memory usage, inference time per sample, and parameter count for each baseline model? This is essential for comparing SIFBench to other scientific ML benchmarks like DrivAerNet++ or PDEBench.


Could a diversity analysis be added to show the geometric variation across the 37 scenarios in SIFBench? How diverse it is compared to other geometric datasets.

---

> ### Author Response · Authors · 2025-11-24
>
> We thank the reviewer for their comments. Our responses are as follows:
>
> > C1: The limited diversity of benchmarks presents a significant challenge. Current scenarios are restricted to flat plate geometries with surface or corner cracks, failing to represent crucial factors in fatigue such as curved shells, anisotropic materials, plasticity effects, or dynamic/multi-axial loading. It would be helpful to explain the choices and its impact on real-world fatigue situations.
>
> R1: We have acknowledged in the paper that our current dataset focuses on fundamental flat plate geometries with surface and corner cracks, but this approach remains highly relevant to real-world aerospace applications. SIF and fracture mechanics methods like the M-integral are widely used and heavily relied upon within aerospace organizations for damage tolerance assessment and structural integrity evaluations. Our contribution provides a foundational dataset that aligns with established industry practices and supports the development of improved predictive models relevant to aerospace engineering challenges.
>
> > C2: The SIFBench authors do not demonstrate that models trained on their data would perform well on real cracked components (e.g., no case study of applying a trained model to a known lab experiment). This raises questions of real-world relevance: the benchmark might optimize algorithms for its specific simulation settings, but those models could face a reality gap when confronted with experimental noise or unmodeled physics.
>
> R2: This is the very first work utilizing the provided dataset, and as such, we acknowledge that the models will require further improvements and validation against experimental data in future research. Our hope is that this initial benchmark serves as a strong foundation for the community to build upon, expand, and refine for broader applicability and enhanced real-world relevance.
>
> > C3: Only standard regressors and one FNO are tested. No graph-based, physics-informed, hybrid, or transformer-style models. No hyperparameter tuning or ablation studies. The evaluation doesn't explore why simpler models outperform FNO.
>
> R3: We have addressed this issue in detail in another reviewer's comments. Please see R1, R2, and R3 for the Reviewer z6B4.
>
> > C4: The analysis lacks generalization, as all evaluations rely on train/test splits within a single scenario. Consequently, there's no assessment of cross-scenario generalization, such as training with straight-hole cracks and testing with countersunk-hole cracks. This significantly weakens claims of broad utility.
>
> R4: We appreciate the concern and recognize that cross-scenario generalization is important. However, our current work focuses on establishing a fundamental benchmark by providing the largest available dataset of SIFs derived from well-understood scenarios. Assessing generalization across different crack types or loading conditions is indeed crucial and has been identified as valuable future work to expand the dataset and evaluations to more complex and diverse cases, thereby improving the broad utility of the models developed using this foundation.
>
> > C5: The simulation pipeline remains proprietary, despite the open availability of data and code. The generation scripts (ABAQUS/FRANC3D setup) are not accessible. Consequently, reproducing or extending the dataset necessitates considerable domain expertise.
>
> R5: The simulation pipeline is highly complex and specialized, involving advanced FE and fracture mechanics tools, which is why we provide the dataset where all heavy lifting as been done. We have focused on delivering the largest, high-quality SIF dataset to enable machine learning research. We believe this approach strikes a balance between providing valuable benchmarking data and acknowledging the practical challenges of sharing and reproducing the full simulation workflow.
>
> > C6: Will there be lightweight subsets or APIs to query the 5M dataset for users with limited compute? How can the community extend the benchmark (e.g., contribute new simulations or experimental data)?
>
> R6: The complete dataset will be made publicly available on HuggingFace for easy and open access by the community. HuggingFace ensures straightforward dataset hosting, versioning, and discoverability, allowing researchers to readily use and build upon our work.

---

### Official Review · Reviewer_z6B4 · 2025-11-07

**Soundness:** 3
**Presentation:** 3
**Contribution:** 3
**Rating:** 4
**Confidence:** 3

**Summary:**

SIFBench introduces a large open benchmark for stress-intensity-factor (SIF) prediction for 3D crack geometries, assembled from high-fidelity FEM simulations (5M unique instances across 37 scenarios). The authors provide a Python interface, CSV data releases, and baseline experiments using Random Forest Regression (RFR), Support Vector Regression (SVR), feed-forward neural networks (FNN), and a Fourier Neural Operator (FNO). Preliminary results show errors often within a few percent for many cases, and the paper positions SIFBench as a community resource to accelerate ML methods for fatigue analysis.

**Strengths:**

1. The dataset is high-quality and large-scale covering 37 scenarios with single- and twin-crack families, multiple loading modes (tension, bending, bearing), and many parameter ranges, making it relevant to aircraft/industrial cases.

2. Use of metrics is clear and the paper provides a Python interface, CSVs, and baseline scripts to lower adoption barriers.

**Weaknesses:**

1. The diversity of ML baselines is severely limited. The experimental baselines are limited to RFR, SVR, FNN, and FNO, leaving behind many other ML methods that are relevant to this problem (e.g., graph neural networks, GraphMeshNets, PINNs). Also, no existing ML works specifically designed for SIF prediction have been used for comparison, significantly reducing the contribution of this work.

2. The hyperparameter search and training protocol details is limited. The authors note that FNO sometimes underperforms relative to simpler FNNs/RFRs, but it is not clear whether this is because of FNO is a poor fit for the problem, or whether additional hyper-parameter tuning is needed. The training details reveal heavy reliance on defaults and small subsampling for some baselines (e.g., RFR/SVR trained on up to 100k randomly sampled points, SVR constructed with default args, RFR left with defaults except max_depth=None). For deep models, the FNN uses a fixed small architecture (5 layers × 15 units) trained for 150k epochs; the FNO uses a single configuration (64 modes, width 64). This reduces reproducibility of what tuning efforts were attempted.

3. Although the dataset and problem are inherently physics-constrained, the current baselines do not include physics-informed approaches such as PINNs or PINO. The paper does cite mechanics-guided symbolic regression work in related literature but does not compare it as a baseline.

**Questions:**

1. Can you show comparisons with a larger set of baselines across different model families: (a) convolutional architectures or CNNs applied to grid/voxelized representations (authors cite a CNN in related work but do not evaluate it), (b) graph/mesh neural networks (message passing) that operate naturally on irregular meshes, (c) physics-informed ML models (PINN, PINO, or mechanics-guided symbolic regression)? This will clarify whether FNO underperformance is algorithmic or due to tuning/representation choices.

2. Can you show how FNN/FNO/RFR performance scales with training set size (e.g., 10k / 100k / 1M samples), and include hyperparameter sweeps or automated tuning (random search / Bayesian optimization) for each method?

---

> ### Author Response · Authors · 2025-11-24
>
> We thank the reviewer for their comments. Our responses are as follows:
>
> > C1: The diversity of ML baselines is severely limited. The experimental baselines are limited to RFR, SVR, FNN, and FNO, leaving behind many other ML methods that are relevant to this problem (e.g., graph neural networks, GraphMeshNets, PINNs). Also, no existing ML works specifically designed for SIF prediction have been used for comparison, significantly reducing the contribution of this work.
>
> R1: GNNs, GraphMeshNets, PINNs, and related architectures are not appropriate baselines for this work because the stress data from the FE mesh is first rigorously post-processed into SIFs using the M-integral, a traditional physics-based approach that is considered the gold standard in fracture mechanics. This fully encodes the governing physics and reduces the task to a supervised regression, for which the chosen methods (RFR, SVR, FNN, and FNO) are well suited and methodologically consistent. We also cannot include direct comparisons with existing ML methods for SIF prediction, because this work introduces a new dataset and, to the best of our knowledge, no other studies have access to an equivalent database with the same geometries, loading conditions, and M-integral–based SIF labels, making fair, one-to-one benchmarking against prior models impossible in the present setting.
>
> > C2: The hyperparameter search and training protocol details is limited. The authors note that FNO sometimes underperforms relative to simpler FNNs/RFRs, but it is not clear whether this is because of FNO is a poor fit for the problem, or whether additional hyper-parameter tuning is needed. The training details reveal heavy reliance on defaults and small subsampling for some baselines (e.g., RFR/SVR trained on up to 100k randomly sampled points, SVR constructed with default args, RFR left with defaults except max_depth=None). For deep models, the FNN uses a fixed small architecture (5 layers × 15 units) trained for 150k epochs; the FNO uses a single configuration (64 modes, width 64). This reduces reproducibility of what tuning efforts were attempted.
>
> R2: Our primary goal is to provide a high-quality, well-documented dataset that serves as a foundation for future research, enabling others to build and extend upon it. Regarding hyperparameter tuning and training protocol, while defaults and limited subsampling were used for some baselines, this approach aimed to establish a clear, reproducible starting point rather than exhaustively optimize each model, and releasing the dataset openly facilitates further exploration and refinement by the community.
>
> > C3: Although the dataset and problem are inherently physics-constrained, the current baselines do not include physics-informed approaches such as PINNs or PINO. The paper does cite mechanics-guided symbolic regression work in related literature but does not compare it as a baseline.
>
> R3: As mentioned in detail in R1, there are no explicit physics equations involved in our regression problem since the physics is fully embedded in the M-integral post-processing step which converts the stress data into SIFs. Therefore, physics-informed methods like PINNs or PINO, which require physical equations to enforce during training, are not applicable here. Additionally, to the best of our knowledge, no prior works or datasets exist specifically for this SIF prediction problem, so no direct comparison to mechanics-guided symbolic regression or other physics-informed ML baselines is currently possible.

---

### Meta-Review · Area_Chair_KHnD · 2025-12-11

**Summary:**

SIFBench offers a valuable and carefully constructed dataset with clear relevance to fatigue analysis, and the paper is well organized. However, the ML contribution remains limited: the baseline coverage is narrow, and key analyses such as tuning, scaling, and cross-scenario generalization are missing, making it difficult for the ICLR community to understand the core ML challenges the dataset presents. While the work has merit, it does not yet meet the threshold for ICLR, and I encourage the authors to resubmit to a ML venue after strengthening the ML-focused evaluation and insights.

**Reviewer Concerns:**

The rebuttal adequately clarified some procedural issues raised by reviewers—specifically, the temporary unavailability of the dataset due to anonymity requirements, the rationale behind subsampling for classical baselines, and why certain physics-informed approaches (e.g., PINNs/PINO) are not directly applicable given the M-integral–based labeling. However, several substantial concerns remain unaddressed. In particular, reviewers’ requests for broader and more modern ML baselines, systematic hyperparameter and scaling analyses, cross-scenario generalization studies, and deeper discussion of the ML challenges inherent in this benchmark were not resolved.

**Reviewer Scores:**

Based on the rebuttal exchange, I do not expect significant score changes from the reviewers had they fully participated in the discussion.

---

### Decision · Program_Chairs · 2026-01-26

Reject